# Molecular basis of TMPRSS2 recognition by *Paeniclostridium sordellii* hemorrhagic toxin

Ruoyu Zhou[1,2,3,4,6], Liuqing He [2,3,4,5,6], Jiahao Zhang[2,3,4,6], Xiaofeng Zhang [2,3,4], Yanyan Li[2,3,4], Xiechao Zhan [2,3,4] & Liang Tao [1,2,3,4,5]

Hemorrhagic toxin (TcsH) is a major virulence factor produced by *Paeniclostridium sordellii*, which is a non-negligible threat to women undergoing childbirth or abortions. Recently, Transmembrane Serine Protease 2 (TMPRSS2) was identified as a host receptor of TcsH. Here, we show the cryo-EM structures of the TcsH-TMPRSS2 complex and uncover that TcsH binds to the serine protease domain (SPD) of TMPRSS2 through the CROP unit-VI. This receptor binding mode is unique among LCTs. Five top surface loops of TMPRSS2[SPD], which also determine the protease substrate specificity, constitute the structural determinants recognized by TcsH. The binding of TcsH inhibits the proteolytic activity of TMPRSS2, whereas its implication in disease manifestations remains unclear. We further show that mutations selectively disrupting TMPRSS2-binding reduce TcsH toxicity in the intestinal epithelium of the female mice. These findings together shed light on the distinct molecular basis of TcsH-TMPRSS2 interactions, which expands our knowledge of host recognition mechanisms employed by LCTs and provides novel targets for developing therapeutics against *P. sordellii* infections.

*Paeniclostridium sordellii* (also known as *Clostridium sordellii*), is an anaerobic, spore-forming, and gram-positive bacterium commonly found in the soil and the gastrointestinal tracts of animals[1]. In humans, the bacterium opportunistically colonizes the gastrointestinal and genital tracts and releases devastating toxins, resulting in acute symptoms including peritonitis, myonecrosis, gangrene, sepsis, toxic shock syndrome, and fatality[1–3]. Approximately 3–4% of women carry *P. sordellii* and thus are particularly vulnerable to acute infections after gynecologic procedures[4]. It is reported that unsafe abortion leads to septicemia complicated by toxic shock with a mortality of 100%[2].

Hemorrhagic toxin (TcsH, ~300 kDa) and lethal toxin (TcsL, ~270 kDa), which were first described in 1969, are two major virulence factors of *P. sordellii*[5,6]. Both toxins belong to the large clostridial toxin (LCT) family, which is a group of potent exotoxins secreted by

clostridial species with large protein sizes. Among major LCT family members, including *Clostridioides difficile* toxin A (TcdA) and toxin B (TcdB), *P. sordellii* TcsL and TcsH, *Clostridium novyi* alpha-toxin (Tcnα), and *Clostridium perfringens* large cytotoxin (TpeL), TcsH shares the highest homology with TcdA, with a sequence identity of ~77%[7].

Similar to other LCTs, TcsH comprises four structural domains: an N-terminal glucosyltransferase domain (GTD), a cysteine protease domain (CPD), a combined transmembrane delivery and receptor-binding domain (DRBD), and a C-terminal combined repetitive oligopeptides (CROPs) domain (Fig. 1a)[8,9]. These domains together contribute to a multi-step intoxication process: the toxin initially recognizes the receptor(s) of target cells and undergoes endocytosis; low pH inside the endosome

[1]College of Life Sciences, Fudan University, Shanghai 200433, China. [2]Center for Infectious Disease Research, Westlake Laboratory of Life Sciences and Biomedicine, Hangzhou 310024, China. [3]Key Laboratory of Structural Biology of Zhejiang Province, School of Life Sciences, Westlake University, Hangzhou 310024, China. [4]Westlake Institute for Advanced Study, Hangzhou 310024, China. [5]Research Center for Industries of the Future, Westlake University, Hangzhou 310024, China. [6]These authors contributed equally: Ruoyu Zhou, Liuqing He, Jiahao Zhang. ✉e-mail: zhanxiechao@westlake.edu.cn; taoliang@westlake.edu.cn

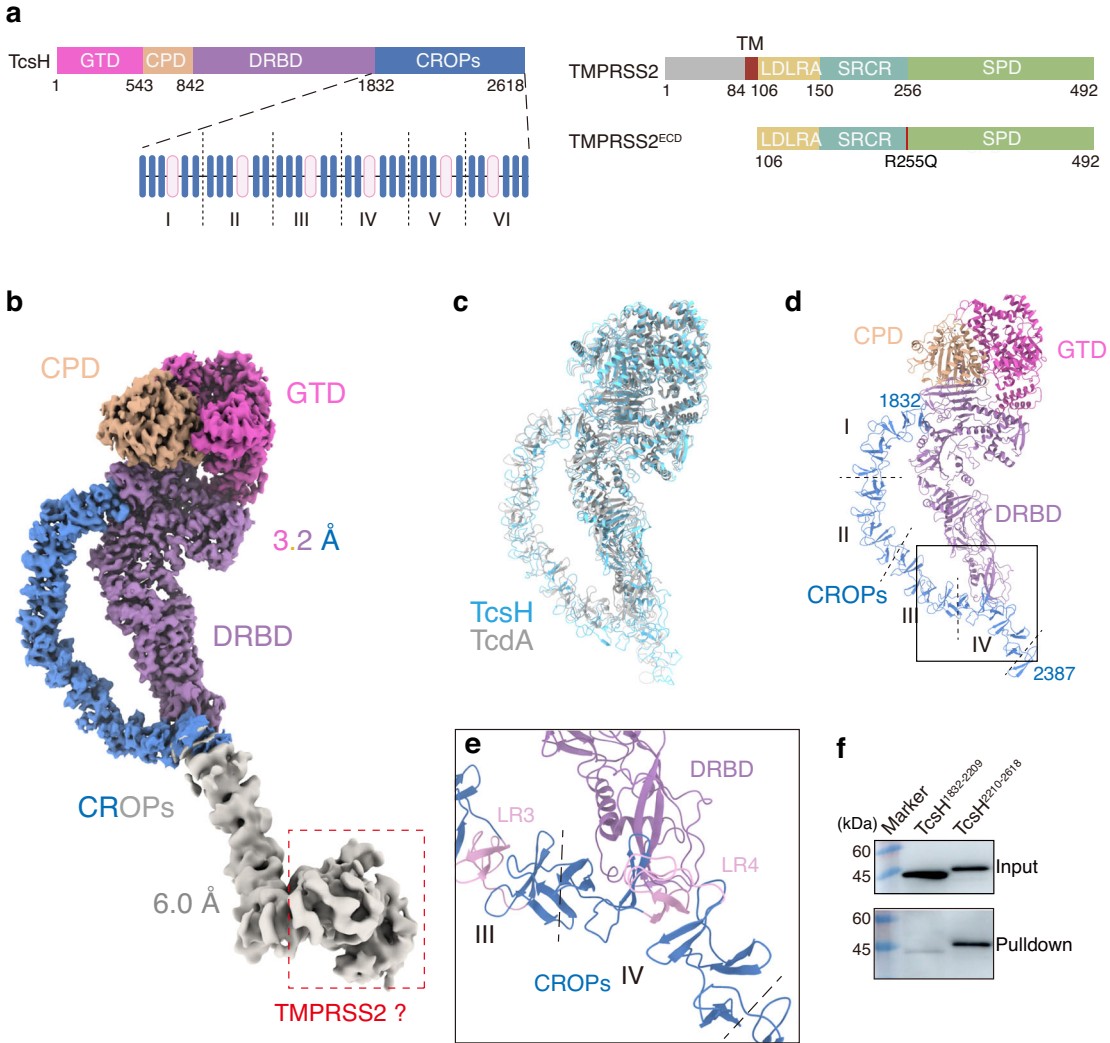

**Fig. 1 | TMPRSS2 binds to the C-terminus of TcsH. a** The schematic diagram showing the domain structures of TcsH and TMPRSS2, as well as the domain boundaries for TcsH[2236-2618] and TMPRSS2[ECD] used for cryo-EM studies. GTD glucosyltransferase domain, CPD cysteine protease domain, DRBD delivery and receptor-binding domain, CROPs combined repetitive oligopeptides domain. The CROP unit-I to VI were marked in the enlarged CROPs domain; blue and pink bars represent SRs (short repeats) and LRs (long repeats) respectively. TM transmembrane domain, LDLRA low-density lipoprotein receptor class A domain, SRCR scavenger receptor cysteine-rich domain, SPD serine protease domain. **b** The EM map of the full-length TcsH in complex with TMPRSS2[ECD] is composed of two parts: a 3.2 Å map for the core region of TcsH holotoxin and colored as shown in Fig. 1a, and a 6.0 Å map for the C-terminal region of TcsH is colored in gray. **c** Superposition of the structure of TcsH[2-2387] (blue) onto a cryo-EM structure of TcdA (gray; PDB: 7POG). **d** Cartoon representation of the core region of TcsH holotoxin. **e** CROP unit-IV bridges to DRBD in the full-length TcsH. LRs are highlighted in pink. **f** The pull-down experiment showed that His-tagged TcsH[2210-2618], but not TcsH[1832-2209], binds to immobilized Fc-TMPRSS2[ECD]. Source data are provided as a Source Data file.

triggers a yet unclear conformational change of the DRBD, which allows the GTD and CPD across the membrane[10,11], followed by a CPD-mediated and Inositol hexakisphosphate (InsP6)-induced autocleavage in the cytosol[12–15]; the released GTD then glucosylates and inactivates Rho GTPases, leading to cytopathic and cytotoxic effects[7,16,17]. The CROPs are special elements exclusively found in the LCT family, which may have carbohydrate-binding capacity[18,19]. For TcsH, the CROPs domain consists of twenty-nine 20 to 24-amino acid short repeats (SRs) interspersed with six 31-amino acid long repeats (LRs).

Researchers have made encouraging progress in resolving the full-length or near-complete structures of LCTs including TcdA[20,21], TcdB[22–25], and TcsL[26]. At neutral pH, TcdA displays in a "closed" conformation that the CROPs domain curves downward alongside the DRBD, while TcdB and TcsL present in an "open" conformation that the CROPs domain curves upward around the GTD-CPD head. Under the low-pH condition, the structures of TcdB and TcsL would convert from

the open to closed conformation[22,26]. Little is known about the full-length structure of TcsH.

Following the recent progress in defining LCT receptors, structures of some LCT-receptor complexes have been resolved, including TcdB1-FZD2[27], TcsL-SEMA6A[28], TcdB1-CSPG4[29], and TcdB4-TFPI[25]. On the other hand, the structural basis of toxin recognition for other receptors, including low-density lipoprotein receptor (LDLR) family proteins for TcdA and Tcnα[30–33], low-density lipoprotein receptor-related protein 1 (LRP1) for TpeL[34], and poliovirus receptor-like 3 (PVRL3) for TcdB[35], remain to be investigated.

Recently, we reported that Transmembrane Serine Protease 2 (TMPRSS2) is a gastrointestinal epithelial receptor for TcsH[36]. TMPRSS2 is a type II transmembrane protein that contains an extracellular domain consisting of a low-density lipoprotein receptor class A (LDLRA) domain, a scavenger receptor cysteine-rich (SRCR) domain, and a serine protease domain (SPD)[37]. TcsH recognizes the SPD of TMPRSS2 (TMPRSS2[SPD]) in a CROPs-dependent manner, but the exact

molecular mechanism is unknown. TcsH also binds to fucosylated glycans on the cell surface, as knocking out key genes for fucosylation, such as *GMDS*, *FUT*, or *SLC3SC1*[38], results in drastically reduced TcsH binding/entry in the MCF-7 cells[36].

Here, we determined the cryo-electron microscopy (cryo-EM) structure of the full-length TcsH in complex with its protein receptor TMPRSS2. We also demonstrate that TcsH binding potently inhibits the proteolytic activity of TMPRSS2, which may lead to potential applications in related biological studies on cancer signaling and anti-viral therapies.

## Results

### Cryo-EM structure of the full-length TcsH
To understand how TcsH recognizes TMPRSS2, we set out to determine the structure of full-length TcsH in complex with the ectodomain of TMPRSS2. For ease of description, we define each LR surrounded by several adjacent SRs as a "CROP unit" (Fig. 1a, Supplementary Fig. 1). Because TMPRSS2 simultaneously undergoes autoproteolysis[39], we introduced an R255Q mutation that prevents the self-cleavage and generated an extracellular domain fragment of TMPRSS2 namely TMPRSS2[106-492/R255Q] or TMPRSS2[ECD] (Fig. 1a). TMPRSS2[ECD] was used for forming the complex with TcsH at neutral pH (Supplementary Fig. 2). The cryo-EM analysis reconstituted a complex map: 3.2 Å for residues 1-2387 including the "core" domains (GTD, CPD, and DRBD) and 6.0 Å for the rest part (Fig. 1b, Supplementary Fig. 3). Notably, an extra lobe of EM density was observed at the C-terminal tip of the CROPs domain and inferred as the bound TMPRSS2[ECD] (Fig. 1b).

The structure of the TcsH part presents a TcdA-like "closed" conformation[20], of which the CROPs domain curves downward alongside the DRBD domain. This structure exhibits an RMSD of about 7.0 Å with TcdA (PDB:7POG) between 2,365 pruned atom pairs (Fig. 1c). The CROPs domain is bridged to the DRBD domain through the CROP unit-IV (Fig. 1d, e), which likely forms a relatively stable configuration of the "core" domains and the CROP unit-I to IV. However, the CROP unit-V and VI along with the presumable TMPRSS2[ECD] were in low resolution, possibly due to the flexibility of these regions in the full-length toxin.

### Structure of the TcsH[2236-2618]-TMPRSS2[ECD] complex
To acquire accurate information regarding the TcsH-TMPRSS2 binding interface, we managed to recruit smaller CROPs fragments for the cryo-EM study. We generated three CROPs constructs: TcsH[1832-2209], TcsH[2210-2618], and TcsH[2236-2618]. The first two are His-tagged proteins and the latter one is fused with an MBP tag. As expected, the pull-down assay showed that TcsH[2210-2618] but not TcsH[1832-2209] robustly binds to Fc-fused TMPRSS2[ECD] (Fig. 1f), supporting the structural observation that TMPRSS2 binds to the C-terminus of the CROPs domain. Using the BLI assay, we confirmed that both TcsH[2210-2618] and TcsH[2236-2618] bound to Fc-TMPRSS2[ECD] with similar affinity ($K_D$ of 0.37 and 0.75 nM, respectively, Supplementary Fig. 4). MBP-fused TcsH[2236-2618] was selected as a substitutive TcsH fragment to form a complex with TMPRSS2[ECD] for the cryo-EM study.

We then determined the cryo-EM structure of the TcsH[2236-2618]-TMPRSS2[ECD] complex, resulting in a density map of intermediate resolution. To represent a more detailed interface, we masked out the marginal parts of the EM-map, and the local resolution of the TcsH-TMPRSS2 interface (residues 2460-2616 for TcsH and residues 142-492 for TMPRSS2) was further improved to 3.0 Å (Supplementary Fig. 5). This structure exactly shows that the SPD of TMPRSS2 binds to the CROP unit-VI of TcsH (Fig. 2a, b and Supplementary Table 1).

### Structure of the TcsH-TMPRSS2[ECD] complex
The newly determined TcsH[2236-2618]-TMPRSS2[ECD] complex map together with the earlier 3.2 Å EM map for TcsH allows us to reconstitute the atomic model of the TcsH-TMPRSS2 complex (Fig. 2c, Supplementary

Fig. 6). This practice also generates a complete TcsH CROPs domain from unit-I to VI, exhibiting a long curved β-solenoid fold pinched at LR4 with each LR/SR consisting of a single β-hairpin followed by a loop (Fig. 2d).

### TcsH recognizes TMPRSS2 through interactive networks
The TMPRSS2[ECD] adopts a conserved chymotrypsin/trypsin fold with two six-strand β barrels and a catalytic center containing the canonical Ser441-His296-Asp345 catalytic triad. This structure is almost identical to the previously reported one[40] (PDB: 7MEQ) with an RMSD of 1.025 Å between 322 pruned atom pairs (Supplementary Fig. 7a). The top surface of TMPRSS2[SPD] contains eight surface loops, namely LA, LB, LC, LD, LE, L1, L2, and L3 (Supplementary Fig. 7b), which determine the substrate specificity of the protease[41]. In the TcsH-TMPRSS2[ECD] complex, LA, LB, LE, L2, and L3 from TMPRSS2 together hold the CROP unit-VI of TcsH (Fig. 3a), like a big hand grabbing a guinea pig from the top-view (Fig. 3b). Interactive networks are observed predominantly through hydrophobic interactions and a few hydrogen bonds with a buried interface at about 1130 Å² (Fig. 3c–e). Particularly, side chains of H2489 from SR25, E2544 from LR6, and Q2520 from SR26 form three hydrogen bonds with Y322 from LE and Q276 from LA (Fig. 3c). L2558 from SR27 and F2605/I2608 from SR29, together with V280 from LA, L302 from LB, L419 from L3, and W461 from L2 contribute to extensive hydrophobic interactions (Fig. 3d, e). In addition, the backbone amino groups of G2559 from SR27 and F2596 from SR29 form two hydrogen bonds with E299 from LB and S463 from L2 of TMPRSS2, respectively (Fig. 3d, e).

### Validation of TcsH-TMPRSS2 interactions by site-specific mutagenesis
To confirm the TcsH-TMPRSS2 interface, we performed structure-guided mutagenesis on TMPRSS2[ECD]. Site-directed point mutations on different loops, including V275R and Q276 A at LA, L302R and H307A at LB, and D417A and L419R at L3, all attenuate the TcsH-TMPRSS2 interaction, as demonstrated by the pull-down assay (Fig. 4a). We also generated several mutations on TcsH[2210-2618] according to the structure. Some of them, such as Q2520A/F2521A, Q2520F/F2521R, E2544R, and F2557N/L2558N, substantially impair the binding capability of TcsH[2210-2618], as demonstrated by the pull-down assay (Fig. 4b). Consistently, these TcsH[2210-2618] mutants also showed reduced binding to the surface of MCF7 *GMDS*[−/−] cells (Fig. 4c), which expresses TMPRSS2 but not fucosylated glycans. We further introduced mutations, including Q2520A/F2521A, E2544R, F2557N/L2558N, and Q2520A/F2521A/E2544R into the full-length TcsH. The circular dichroism (CD) spectra analysis showed that all these mutant toxins, including TcsH[Q2520A/F2521A], TcsH[E2544R], TcsH[F2557N/L2558N], and TcsH[Q2520A/F2521A/E2544R] (referred to as TcsH[AAR] thereafter), were properly folded (Supplementary Fig. 8). When applied to MCF7 cells, all the mutants were less toxic than TcsH (~8 to 50-fold) to the WT cells (Fig. 4d) but equally potent to the *TMPRSS2*[−/−] cells (Fig. 4e), indicating that these designed mutations specifically impair the TMPRSS2 recognition.

Albeit the TcsH CROP unit-VI and its homologous fragment in TcdA (CROP unit-VII) are sequentially and structurally close[42] (Supplementary Fig. 9), it was reported that the TcdA CROPs domain does not bind to TMPRSS2[36]. By interrogating the different residues between the TcsH CROP unit-VI and TcdA CROP unit-VII, we identified that F2521I, a single-point substitution in TcsH, is capable of abolishing the TMPRSS2 binding of TcsH[2210-2618] where some other mutations also partly affect the interaction (Supplementary Fig. 10a). On the other hand, reversely mutating the according residue in TcdA (residue position 2613) from I to F did not render the TcdA CROPs overt binding to TMPRSS2 (Supplementary Fig. 10b).

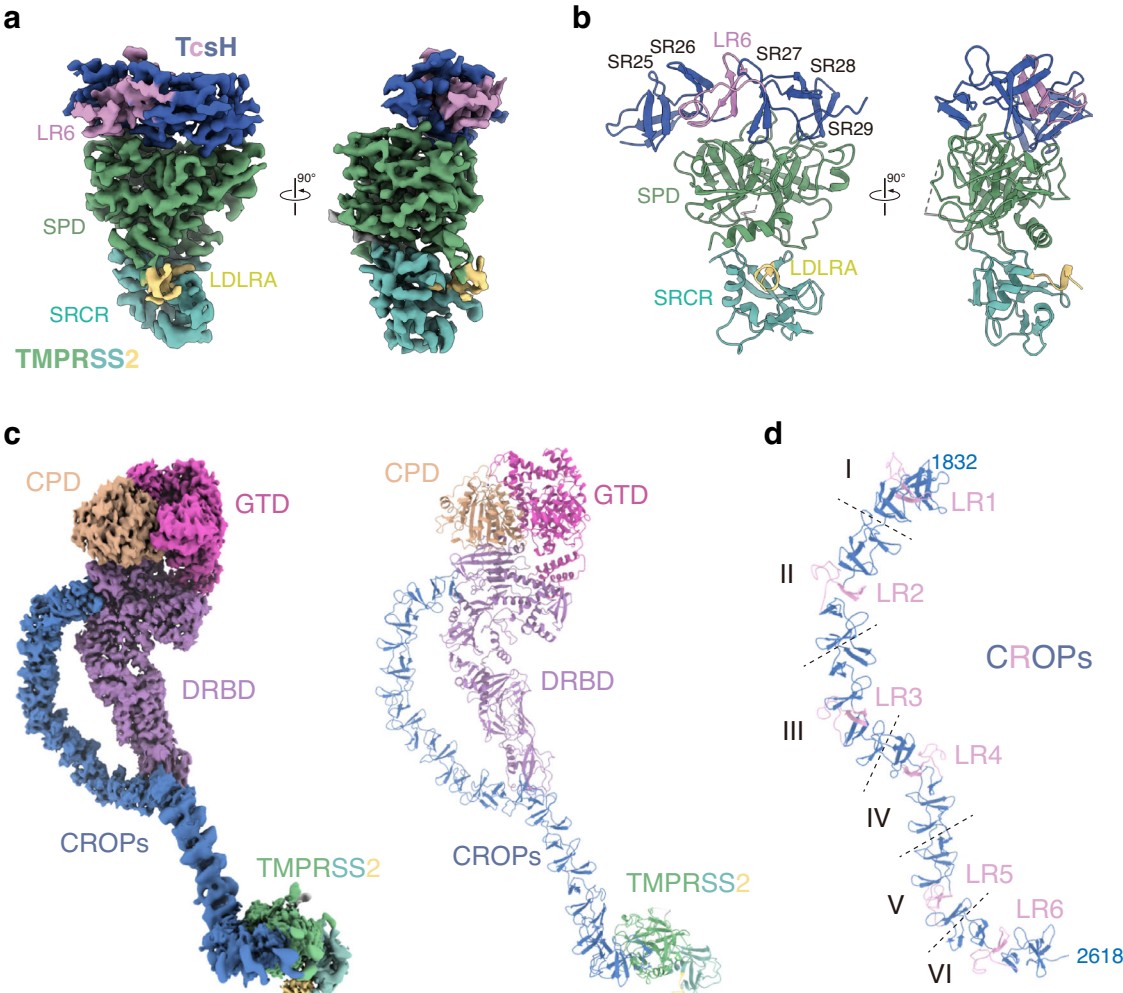

**Fig. 2 | Overall structure of TcsH-TMPRSS2^ECD complex. a** The 2.95 Å resolution cryo-EM map of the TcsH^2236-2618-TMPRSS2^ECD complex in two views and colored as shown in Fig. 1a. **b** Cartoon representation of the structure of the TcsH^2460-2616-TMPRSS2^ECD complex that is shown in similar orientations and color schemes as that in (**a**). **c** Left panel: EM map of the complete TcsH-TMPRSS2^ECD complex; right panel: cartoon representation of the complete TcsH-TMPRSS2^ECD complex. **d** Ribbon diagram showing the structure of the CROPs domain with LR1-6 colored in pink. The entire CROPs domain is divided into six CROP units by dashed lines.

## TcsH binding inhibits the proteolytic activity of TMPRSS2

The proteolytic activity of TMPRSS2 was achieved by the catalytic triad composed of S441-H296-D345[43] (Fig. 5a). Based on the structure of TcsH-TMPRSS2^ECD complex, the TcsH CROP unit-VI largely covered the flattop surface of TMPRSS2^SPD, with the β-hairpin of SR27 and the loop of SR29 stretching into the catalytic pocket (Fig. 5b, Supplementary Fig. 11a).

We then tested whether the binding of TcsH inhibits its proteolytic activity of TMPRSS2. A fluorogenic peptide Boc-Gln-Ala-Arg-7-amino-4-methylcoumarin (Boc-QAR-AMC) was used as a reporting substrate for the proteolytic activity of TMPRSS2[44]. Active TMPRSS2 can cleave the peptide right after Arg thus the fluorescent signals are emitted (Fig. 5c). We showed that the proteolytic activity of TMPRSS2 could be well suppressed by the addition of TcsH^2210-2618 (Fig. 5d) but not TcsH^2210-2618 mutants lacking TMPRSS2-binding ability (Fig. 5e). These results suggest that the TcsH CROPs-derived fragments may serve as potent protease inhibitors specific for TMPRSS2.

We also tested three serine protease inhibitors of TMPRSS2, including Bromhexine, Nafamostat, and Avoralstat. Nafamostat modifies Ser441 on TMPRSS2[40], Bromhexine, and Avoralstat may also target the catalytic triad of TMPRSS2 in the same way[45]. However, all three chemicals failed to protect cells from TcsH (Supplementary Fig. 11b), likely because the modification moiety in the catalytic triad does not interfere with the toxin-receptor binding interface (Supplementary Fig. 11c).

## TcsH mutant with impaired TMPRSS2 binding causes less epithelial damages

Lastly, we analyzed the toxicity of a TMPRSS2-binding defective mutant (TcsH^AAR) in comparison with the WT toxin by injecting them into the ligated mouse colons. The dissected colon tissues were processed and stained using hematoxylin and eosin (H&E) to examine the toxin-induced damage in the intestinal epithelium. We observed that the WT TcsH induced overt epithelial damage to the colon tissue, resulting in inflammatory cell infiltration, epithelial disruption, and hemorrhage. TcsH^AAR showed attenuated potency and caused greatly reduced epithelial disruption and hemorrhage to the colonic epithelium (Fig. 6). These results further confirmed our TMPRSS2-TcsH structure as well as the role of TMPRSS2 in TcsH-induced colonic epithelial lesions.

## Discussion

The host receptor is a primary determinant of the specificity and efficacy of a toxin that further affects the manifestations of related bacterial infection-associated diseases. Recently,

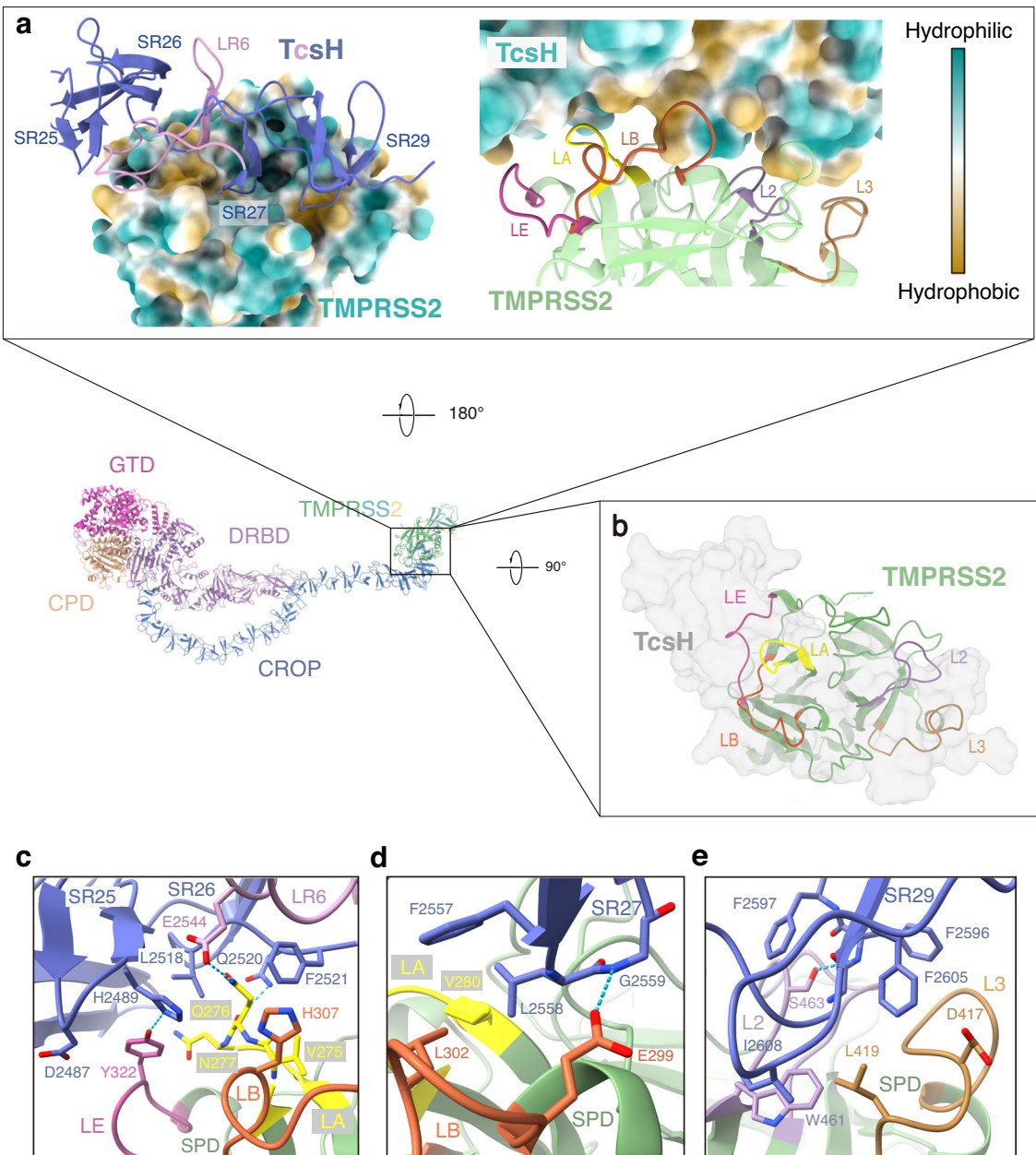

**Fig. 3 | TcsH-CROP unit-VI recognizes TMPRSS2^SPD. a** Hydrophobicity of the TcsH-TMPRSS2 interface. The SRs and LRs of TcsH and surface loops of TMPRSS2 that participate in the interactions are colored and labeled as follows: LR: pink; SR: dark blue; LE: magenta; LA: yellow; LB: orange; L2: purple; L3: brown; (**b**) The top view of the complex shows that TMPRSS2^SPD interacts with TcsH through five loops. **c**–**e** Close-up views of the TcsH-TMPRSS2 interface with interacting side chains shown as sticks and colored as shown in (**a**). Hydrogen bonds are shown as blue dotted lines.

TMPRSS2 and fucosylated glycans were identified as cellular receptors for TcsH, a major toxin produced by *P. sordellii*[36]. Here, we report the cryo-EM structures of the full-length TcsH and the partial CROPs in complex with the ectodomain of TMPRSS2, revealing the underlying mechanism of host recognition by this notorious toxin (Supplementary Fig. 12).

We show that TcsH binds to TMPRSS2 through the end of the CROPs domain, or more precisely the CROP unit-VI. Whereas the CROPs domains of LCTs are generally thought to bind sugar moieties on the cell surface, TMPRSS2 is the first protein receptor reported that solely binds to the CROPs in this toxin family. Each LCT recognizes distinct host receptors, while two types of protein receptor-binding interfaces were well-characterized before. The first one is located at the convex edge of the DRBD, which is thought to be evolutionarily developed for receptor recognition[46]. Several LCT receptors bind to

this region, including Frizzled proteins (FZDs) for TcdB1 and TcdB3[27,47], Semaphorin 6A and 6B (SEMA6A/6B) for TcsL[28,48], and tissue factor pathway inhibitor (TFPI) for TcdB2 and TcdB4[25,49]. The second interface lies in an area where CPD, DRBD, and CROPs converge; chondroitin sulfate proteoglycan 4 (CSPG4) for TcdB is the only reported receptor that binds to this region[29,50]. Our finding of the C-terminus of CROPs as a new protein receptor-binding region for LCTs is intriguing but also puzzling: while a distinct receptor-binding domain can usually be defined in most toxins, LCTs seem to have various receptor-binding interfaces/regions. Nevertheless, the presence of multiple receptor-binding regions potentiates LCTs to explore a broader spectrum of host recognition.

Recent studies have resolved the high-resolution structures of full-length TcdB[22,24,25], full-length TcsL[26], and near-complete TcdA[20,21]. Despite these important achievements, architectural insights into

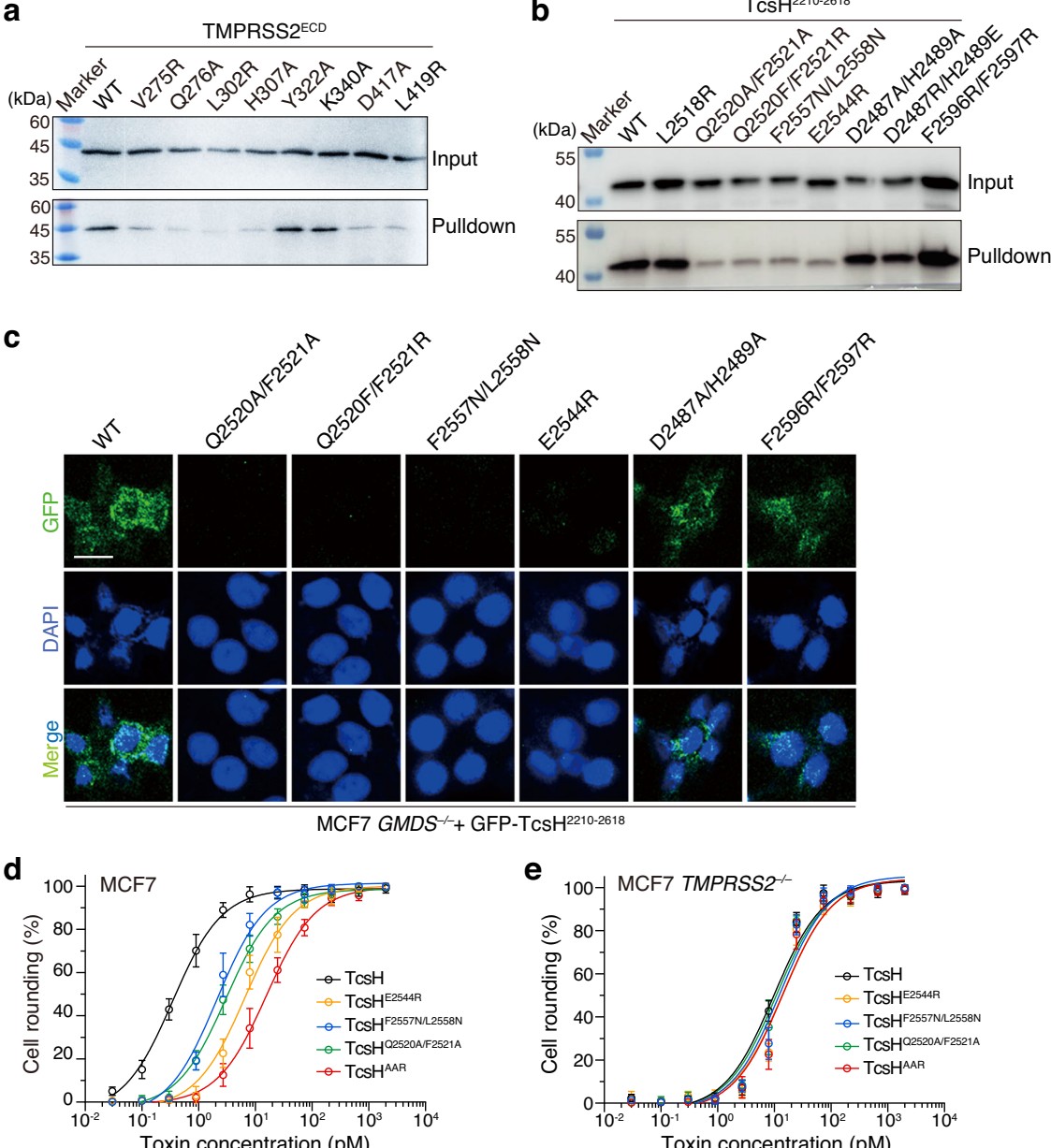

**Fig. 4 | Characterization of the interactions between TcsH and TMPRSS2 by structure-based mutagenesis. a** Binding of the wild-type TcsH[2210-2618] to Fc-His-TMPRSS2[ECD] variants immobilized on Protein A resins was examined using pull-down assays. **b** The binding of TcsH[2210-2618] variants to Fc-His-TMPRSS2[ECD] immobilized on Protein A resins was examined using pull-down assays. **c** Confocal fluorescence images show the binding of different GFP-fused TcsH[2210-2618] variants to the MCF-7 *GMDS*[−/−] cells. Cell nuclei were stained by DAPI (blue). The blots and images are representatives of three independent experiments. The scale bar represents 50 μm. **d**, **e** The sensitivities of the MCF-7 *GMDS*[−/−] and *TMPRSS2*[−/−] cells to TcsH or its mutants were tested using the cytopathic cell-rounding assays. The percentages of the rounded cells were plotted over the toxin concentrations. Error bars ($n = 5$ biologically independent samples) indicate mean ± SD. Source data are provided as a Source Data file.

other LCT members are still lacking. Our current structure of the TcsH-TMPRSS2 complex provides a snapshot of TcsH. Since the TMPRSS2 binding site in TcsH is far away from the core domains and would unlikely cause conformational change to the toxin, we believe that it could represent the structure of native TcsH at neutral pH. The overall structure and conformation of TcsH resemble that of TcdA, indicating the evolutionarily close relationship and homology between these two toxins. TcdA and TcsH contain the longest CROPs among LCTs for unknown reasons. Notably, the reported TcdA structures miss the C-terminal part of the CROPs[20,21], likely due to the high flexibility. Hence, the structure of complete TcsH CROPs may be helpful to study the architecture and orientation of the TcdA CROPs domain. The

C-terminal regions beyond the CROP unit-IV hang freely in the full-length structures of TcdA and TcsH, which seem to be functionally mysterious. Our structural study now provides a rational explanation for these extra-long CROPs: the extra CROP units may serve as variable platforms to explore host recognition. Besides, it would be interesting to inspect whether TcdA also adopts these CROP units to bind uncharacterized receptor(s).

TMPRSS2 recognizes TcsH via multiple loops on the top surface of the SPD, which are also responsible for the recognition of its intrinsic substrates. These loops, including LA, LB, LE, L2, and L3, are highly variable among the TTSP family members[51], explaining why TMPRSS2 is specifically recognized by TcsH. On

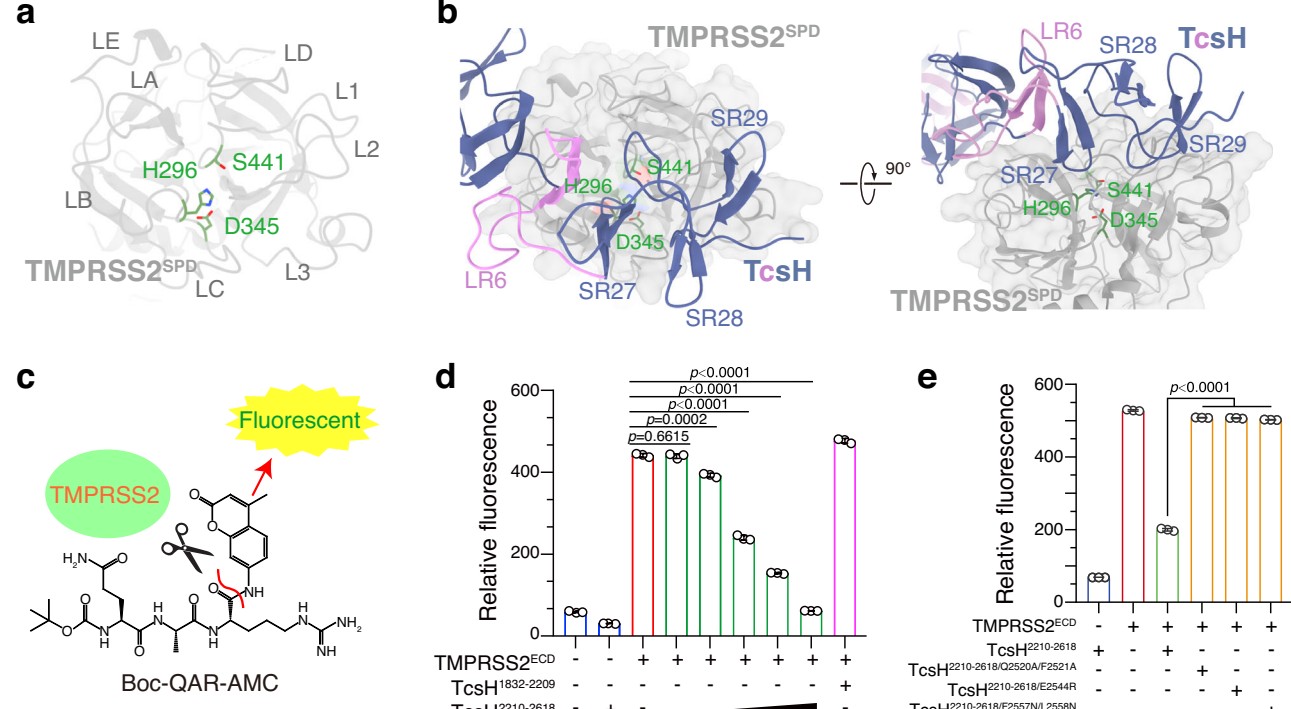

**Fig. 5 | TcsH binding blocks the proteolytic activity of TMPRSS2. a** The top view of TMPRSS2$^{SPD}$ structure with the catalytic triad (H296, D345, S441) in the serine protease domain highlighted with sticks and green color. **b** In the TcsH-TMPRSS2 complex, CROP unit-VI covers the catalytic pocket of TMPRSS2. Loops of SR27 and SR29 stretch into the catalytic pocket and possibly block the substrate entrance. **c** Schematic diagram illustrating the principle of TMPRSS2 protease activity assay. TMPRSS2 cleaves at the bond after the Arginine of a fluorogenic peptide substrate Boc-QAR-AMC, releasing AMC with bright fluorescence emitted. **d** TcsH$^{2210-2618}$ blocks the proteolytic activity of TMPRSS2$^{ECD}$ in a concentration-dependent

manner. The relative fluorescence value (proportional to the proteolytic activity of TMPRSS2) generated from released AMC is shown in a bar chart. The concentration of the substrate, TMPRSS2$^{ECD}$, and TcsH$^{1832-2209}$ is 10 μM, 30 nM, and 2 μM, respectively. TcsH$^{2210-2618}$ concentration gradient: 3.2 nM, 16 nM, 80 nM, 400 nM, and 2 μM. **e** TcsH$^{2210-2618}$ mutants show the attenuated ability to block the proteolytic activity of TMPRSS2$^{ECD}$. The concentrations of Boc-QAR-AMC, TMPRSS2$^{ECD}$, and TcsH$^{2210-2618}$ fragments are 10 μM, 30 nM, and 400 nM, respectively. In (**d**, **e**), data are shown as mean ± SD, $n = 3$ independent samples, two-tailed Student's $t$ test. Source data are provided as a Source Data file.

the other hand, the CROPs of LCTs are also less conserved, implying that this toxin-receptor recognition might still undergo rapid evolutionary adaptation. TMPRSS2 is mainly expressed in the epithelium of the prostate, gastrointestinal tracts, kidney, and pancreas[39,52]. Dysregulated TMPRSS2 activity is related to the proliferation, invasiveness, and metastasis of prostate tumor cells[53,54]. TMPRSS2 is also well-known for facilitating the cellular entry of several viruses, such as influenza viruses and coronaviruses, through cleaving glycoproteins on the viral envelope to activate membrane fusion[55–58]. We demonstrate that the TcsH CROP unit-VI tightly binds to the SPD of TMPRSS2 with sub-nanomolar affinity, masks the catalytic pocket in SPD, and inhibits the enzymatic activity of TMPRSS2. Particularly, the CROP unit-VI exhibits a straightforward and well-organized structural configuration, making it amenable to modifications and rational engineering. Therefore, we also propose that the TcsH CROP unit-VI could be a potential precursor to develop anti-cancer and anti-viral mediations. Taken together, our study on the molecular basis of recognition between TcsH and its receptor TMPRSS2 would help to understand the bacterial pathogenesis, portray the host-pathogen coevolution, unveil the vulnerability of the devastating toxin, and provide potential therapeutic avenues for the related diseases.

## Methods

### Ethics statement

All animal procedures reported herein were performed following the institutional guidelines and approved by the Institutional Animal Care

and Use Committee at Westlake University (IACUC Protocol #22-018-2-TL). To minimize the distress and pain, the mice injected with toxins were monitored every hour. Animals with signs of pain or distress such as labored breathing, inability to move after gentle stimulation, or disorientation were euthanized immediately. This method was approved by the IACUC and monitored by a qualified veterinarian.

### Cell lines and antibodies

MCF-7 (HTB-22) cells were originally obtained from ATCC and Expi293F cells were purchased from ThermoFisher Scientific (U.S.). MCF-7 *TMPRSS2*$^{−/−}$ and *GMDS*$^{−/−}$ cells were previously generated laboratory stocks[36]. Expi293F cells were cultured in SMM 293-T II Expression Medium (Sino Biological, Beijing, China) under 95% air and 5% CO$_2$ in a Multitron-Pro shaker (Infors) at 37 °C. MCF-7 cells were cultured in DMEM media plus 10% fetal bovine serum and 1% penicillin-streptomycin in a humidified atmosphere of 95% air and 5% CO$_2$ at 37 °C.

The following antibodies were purchased from commercial vendors: mouse monoclonal anti-6×His tag antibody (Proteintech, 66005-1-Ig, 1:5000), goat monoclonal anti-human IgG-Fc antibody (Sino Biological, SSA001, 1:10000), and horseradish peroxidase-labeled goat monoclonal anti-mouse IgG antibody (H + L, PI-1000, Vector Labs, 1:10000).

### Mice

C57BL/6 mice were purchased from the Laboratory Animal Resources Center at Westlake University (Hangzhou, China). Female, 6–8 weeks C57BL/6 mice were used in this study. Mice were housed in specific-

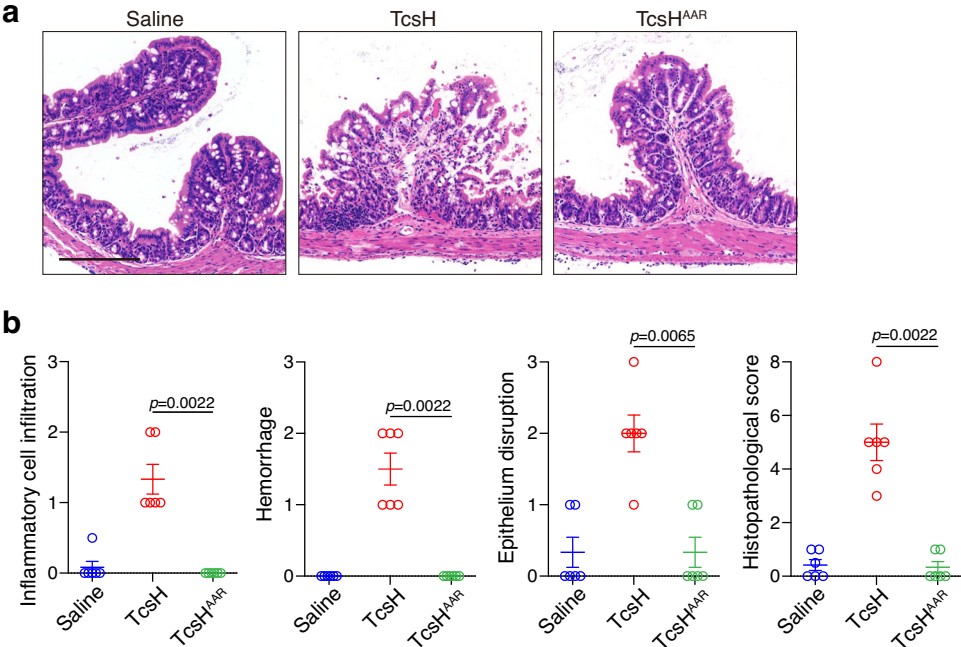

**Fig. 6 | TcsH^AAR causes less intestinal epithelial damage. a** H&E-stained mouse colonic tissue sections harvested after colon-loop ligation assays. Representative images are shown. The scale bar represents 200 μm. **b** Histological scores for (a) were assessed based on inflammatory cell infiltration, hemorrhage, epithelial disruption, and overall. $N = 6$ mice per group, error bars indicate mean ± SEM, Mann–Whitney test. Source data are provided as a Source Data file.

pathogen-free micro-isolator cages with free access to drinking water and food and monitored under the care of full-time staff. All mice had a 12-h cycle of light/darkness (7 a.m. to 7 p.m.), housed at 20–24 °C with 40-60% humidity.

### Cloning of DNA constructs and site-directed mutagenesis

DNA encoding full-length of TcsH (reference sequence: *P. sordellii* 9048) was codon-optimized, synthesized by Genscript (Nanjing, China), and inserted into pHT01 via the sites of *Pst*I/*Xba*I with an additional C-terminal His-tag. The DNA fragments encoding TcsH^1832-2246, TcsH^2236-2618, TcsH^1832-2209, and TcsH^2210–2618 were PCR amplified and inserted into pET28a with a His-tag at the C-terminus. For cryo-EM sample preparation, TcsH^2236-2618 was cloned into a pET21a vector with a His-MBP tag introduced at the N-terminus. The gene encoding the ectodomain of TMPRSS2 (amino acids 106-492) was PCR amplified from the cDNA library and inserted into the AgeI/XhoI sites of a pHLsec vector with an Fc-His, or His-tag fused to their N-terminus. The point mutations of TMPRSS2 and TcsH were generated using QuickChange II Site-Directed Mutagenesis Kit (Agilent Technologies, 200523) or Q5 Site-Directed Mutagenesis Kit (New England Biolabs, E0554S) following the manufacturer's protocol.

### Expression and purification of recombinant proteins

The recombinant full-length TcsH was expressed in *Bacillus subtilis* SL401 as described previously[59]. The fresh transformants were inoculated into 5 mL 2×YT medium supplemented with 50 μg/mL chloramphenicol and allowed to grow overnight at 37 °C. The overnight culture was transferred to one liter of 2×YT medium containing 50 μg/mL chloramphenicol. The culture was grown at 37 °C until an OD$_{600}$ of 0.8, then the temperature was set up to 16 °C and the cell culture was induced with 1 mM isopropyl β-D-1-thiogalactopyranoside (IPTG) for 36 h. Cells were thawed and resuspended in lysis buffer (50 mM Tris-HCl, pH 7.4, 300 mM NaCl, 10% glycerol, 5 mM imidazole) supplemented with 0.5 mg/mL lysozyme (Macklin, L6051), 0.5% Triton X-100 (Sigma, T9284), 5 units/mL DNAse I (Biorigin, BN20219), and 1 μg/mL

RNAse A (ZF-50-0002, Multi Sciences). After rocking for 2 hours at room temperature, the insoluble fraction was removed by centrifugation at 20,000 *g* for 1 hour at 4 °C. The soluble fraction was applied to TALON superflow metal affinity resin (Takara, 635507) and washed with lysis buffer supplemented with 10 mM imidazole (Vetec, V900153). The protein was eluted with a buffer of 20 mM Tris-HCl, pH 7.4, 150 mM NaCl (Buffer A), and 100 mM imidazole. Then, the protein was further purified by size-exclusion chromatography on a Superdex 200 increase 10/300 GL column (GE Healthcare) in Buffer A. Protein fractions were confirmed by SDS-PAGE, concentrated, and stored at −80 °C.

TcsH CROPs fragments were expressed in *E. coli* BL21(DE3) and purified as His- or MBP-tagged proteins. *E. coli* was grown in the Luria-Bertani medium containing 50 μg/mL kanamycin and the protein expression was induced with 0.5 mM IPTG at 16 °C for 16 hours. Bacterial cells were collected by centrifugation, resuspended in Buffer A supplemented with protease inhibitors, and then lysed by passing through an LM20 microfluidizer (Microfluidics) once. Cell lysates were subjected to high-speed centrifugation at 12,000 × *g* for 30 minutes to remove unbroken cells and debris. His-tagged TcsH^1832-2246, TcsH^2236-2618, TcsH^1832-2209, and TcsH^2210–2618 were purified by Ni-affinity chromatography. MBP-tagged TcsH^2236-2618 was purified using Amylose resin (NEB, E8021L) followed by size-exclusion chromatography on a Superdex 200 increase 10/300 GL column (GE Healthcare). Protein fractions were confirmed by SDS-PAGE, concentrated, and stored at −80 °C.

Recombinant TMPRSS2^ECD and its variants were expressed in Expi293F cells and purified as His-tagged proteins. Expi293F cells were transfected with the plasmids when the cell density reached $2 \times 10^6$ cells/mL. Approximately 2 mg of plasmid was preincubated with 4 mg of polyethyleneimine (Polysciences) in 50 mL of fresh medium for 15 minutes. The mixture was added to one liter of cell culture for transfection. After 72 hours, the cell pellets were removed by centrifugation at 4000 × *g* for 10 minutes at 4 °C. The supernatant was collected and concentrated to approximately 200 mL using 10 kDa filters (Sartorius Stedim) and then loaded to a gravity flow column

packed with TALON superflow metal affinity resin. The resin was washed extensively with Buffer A supplemented with 10 mM imidazole and the protein was eluted with Buffer A supplemented with 100 mM imidazole. The eluted protein was concentrated and further applied to size-exclusion chromatography on a Superdex 200 increase 10/300 GL column (GE Healthcare) equilibrated with Buffer A supplemented with 10% glycerol. The target peak fractions were confirmed by SDS-PAGE, then concentrated, and stored at −80 °C.

### Biolayer interferometry (BLI) assay
The binding affinities between toxin fragments and TMPRSS2 proteins were measured using the BLI assay with the Octet RED96e system and analyzed with the Octet Data Analysis software (version 12.0.1.2, ForteBio, Fremont, CA, USA). Briefly, Fc-tagged proteins were immobilized onto the capture biosensors (AHC biosensor, ForteBio) and balanced with binding buffer (20 mM Tris-HCl pH 7.4, 150 mM NaCl). The biosensors were then exposed to the indicated concentrations of MBP-TcsH$^{2236-2618}$ or TcsH$^{2210-2618}$, followed by dissociation in the binding buffer.

### Pull-down assays
Pull-down assays were performed using Protein A agarose beads (Thermo Fisher Scientific). Briefly, Fc-tagged TMPRSS2$^{ECD}$ (50 nM) were mixed with TcsH$^{2210-2618}$ or TcsH$^{1832-2209}$ (10 nM) in 300 μL of binding buffer (20 mM Tris-HCl pH 7.4, 150 mM NaCl). The mixtures were incubated at 4 °C for 2 hours and co-precipitated by Protein A agarose beads. Beads were washed, pelleted, boiled in SDS sample buffer, and subjected to SDS-PAGE or immunoblot analysis.

### Assembly of the TcsH-TMPRSS2 complex
To assemble the TcsH-TMPRSS2$^{ECD}$ and TcsH$^{2236-2618}$-TMPRSS2$^{ECD}$ complexes, purified full-length TcsH or TcsH$^{2236-2618}$ were incubated with TMPRSS2$^{ECD}$ in the presence of 10% glycerol on ice for one hour in molar ratios of 1:5 and 1:1, respectively. These protein complexes were further purified by size-exclusion chromatography on a Superdex 200 increase 10/300 GL column (GE Healthcare) in a buffer containing 20 mM Tris-HCl, pH 7.4, 150 mM NaCl, and 2% glycerol. Protein fractions were confirmed by SDS-PAGE. Peak fractions containing the TcsH-TMPRSS2$^{ECD}$ complex and the TcsH$^{2236-2618}$-TMPRSS2$^{ECD}$ complex were pooled and concentrated to 5 mg/mL and 1 mg/mL for cryo-EM analysis.

### Cryo-EM specimen preparation and data acquisition
For cryo-EM sample preparation, three microliters of each protein were placed on glow-discharged holey carbon grids (Quantifoil Au R2.1/3.1, 300 mesh). The grid was blotted with filter paper for 3.5 s in a chamber set with 100% humidity at 8 °C to remove the excess sample and then plunge-frozen in liquid ethane cooled by liquid nitrogen with the Vitrobot Mark IV system (ThermoFisher Scientific). Cryo-EM specimens were imaged on a 300-kV Titan Krios electron microscope (ThermoFisher Scientific) using a normal magnification of 81,000 rpm. Movies were recorded using a Gatan K3 detector equipped with a GIF Quantum energy filter (slit width 20 eV) at the super-resolution mode, with a physical pixel size of 1.087 Å. Each stack of 32 frames was exposed for 2.56 s, with a dose rate of ~23 counts/second/physical-pixel (~19.5 e-/second/Å$^2$) for each frame using EPU (ThermoFisher Scientific). All 32 frames in each stack were aligned and summed using the whole-image motion correction program MotionCor2[60] and binned to a pixel size of 1.087 Å. The defocus value for each image varied from −1.5 to −2.0 μm and was determined by Gctf[61].

### Cryo-EM data processing
For the TcsH-TMPRSS2$^{ECD}$ complex, a total of 1300 micrographs were collected, of which 1,118 micrographs were selected for further processing. All the processing steps were carried out in cryoSPARC[62] except that especially mentioned. A total of 879,891 particles were extracted with 2× binning (pixel size: 2.174 Å) (Round 1) and subjected to multiple two-dimensional (2D) classifications, resulting in 225,500 good particles. These particles were further classified using Heterogeneous refinement. 148,184 particles from the good class were then re-extracted (pixel size: 1.087 Å) (Round 2) for further 2D classification and Non-uniform refinement, resulting in a final EM reconstruction for the TcsH-TMPRSS2$^{ECD}$ complex at 3.21 Å from 87,378 particles. The 3.21-Å EM density map displays clear features for amino acid side chains in the core region of TcsH. These particles were further locally refined using a soft mask around the C-terminal TcsH and TMPRSS2, which yielded a reconstruction at an average resolution of 5.95 Å.

For the TcsH$^{2236-2618}$-TMPRSS2$^{ECD}$ complex, a total of 3,341 micrographs were collected. A total of 2× binned 8,005,902 particles (pixel size: 2.174 Å) were extracted and applied to multiple rounds of 2D classifications, resulting in 1,975,699 good particles. Followed by Heterogeneous refinement, 887,779 good particles were selected. Further 2D classifications of re-extracted particles (pixel size: 1.087 Å) yielded 760,140 particles. These particles further go through Non-uniform refinement and Local refinement, which generated a reconstruction of the TcsH$^{2236-2618}$-TMPRSS2$^{ECD}$ complex at an average resolution of 2.95 Å.

The reported resolutions were calculated based on the gold-standard Fourier shell correlation (FSC) = 0.143 criteria. Local resolution volumes were estimated using cryoSPARC. The angular distributions of the particles used for the final reconstructions are reasonable. The workflows of cryo-EM data processing are illustrated in Supplementary Fig. 3, 5.

### Model building and refinement
The cryo-EM structure of *C. difficile* toxin A (PDB code: 7POG) was used as a template to generate a homology model for TcsH using CHAINSAW[63]. The homology model was fitted into the cryo-EM map for the TcsH-TMPRSS2$^{ECD}$ complex using UCSF Chimera[64]. Manual adjustment of the model was performed in COOT[65], followed by iterative rounds of real-space refinement in PHENIX[66] and manual adjustment in COOT. Similarly, the crystal structure of the ectodomain of TMPRSS2 (PDB: 7MEQ) was fit into the cryo-EM map and manually adjusted in COOT. The structures were further validated through examination of the Molprobity scores and statistics of the Ramachandran plots. Molprobity scores were calculated as described[67].

### CD spectra analysis of full-length TcsH and mutants
For CD spectroscopy, the sample buffer of the toxin was changed to 10 mM potassium phosphate and 100 mM (NH4)$_2$SO$_4$ (pH 7.4). The protein sample was concentrated to a final concentration of 1 mg/ml for each measurement. CD spectra were recorded using a Chirascan V100 (Applied Photophysics Inc.) in the wavelength range of 190 to 260 nm, with a bandwidth of 1.0 nm and scan step of 0.5 nm using a 0.05-cm path length quartz cuvette at 18 °C. In each case, three spectra were collected, averaged, baseline-corrected, smoothed, and converted with the Chirascan software.

### Cell surface binding assay
MCF-7 *GMDS*$^{-/-}$ cells were incubated with 10 nM WT or mutant GFP-TcsH$^{2210-2618}$ in the medium on ice for 20 min. Cells were washed three times with ice-cold PBS, fixed with 4% paraformaldehyde (PFA) for 15 minutes at room temperature, and stained with DAPI, followed by fluorescence microscopy. Fluorescent images were captured using an Olympus FV3000-BX63 LSCM Confocal System with the software FV31S-SW v2.3.2.169.

## The cytopathic cell-rounding assay

The cytopathic effect of the toxin was analyzed using the cell-rounding assay. Cells were exposed to toxins for 12 hours. The phase-contrast images of the cells were captured by a microscope (Olympus IX73; ×10 objectives) with the software Olympus CellsSens Standard 2.1. Six zones of 200 μm × 200 μm were selected randomly, with each zone containing ~50–150 cells. Round-shaped and normal-shaped cells were counted manually. The percentage of round-shaped cells was analyzed using GraphPad Prism (ver. 9.0.0, GraphPad Software, LLC).

## TMPRSS2 enzymatic activity assay

Peptide substrate N-tert-butoxycarbonyl-Gln-Ala-Arg-7-amino-methylcourarin (Boc-QAR-AMC, Bachem, catalog no. I-1550) was diluted into assay buffer (20 mM Tris-HCl, 150 mM NaCl, 0.1% Triton X-100, pH 8.0) to a final concentration of 10 μM. $TcsH^{2210-2618}$, $TcsH^{1832-2209}$, $TcsH^{2210-2618/Q2520A/F2521A}$, $TcsH^{2210-2618/E2544R}$, $TcsH^{2210-2618/F2557N/L2558N}$, and/or $TMPRSS2^{ECD}$ of indicated concentrations were added into the reaction system with a final volume of 120 μL. After incubation for 2 hours at room temperature, 100 μL of the mixtures were dispensed into 96-well plates (WHB Scientific) and read on a microplate reader (Thermo Varioskan LUX) with fluorescence module at 340 nm excitation and 440 nm emission.

## Colon-loop ligation assay

Six- to eight-week-old female mice were anesthetized by intraperitoneal injection of 1% pentobarbital sodium. A midline-right laparotomy was performed to locate the ascending colon and seal a ~2 cm loop with 4–0 surgical suture ligatures. Six micrograms of TcsH or $TcsH^{AAR}$ in 100 μL of normal saline or 100 μL of saline alone was injected into the sealed colon segment using an insulin syringe, followed by suturing of the skin incision. Mice were allowed to recover in the 37 °C thermostatic plates. After 8 hours, mice were euthanized, and the ligated colon segments were excised. The colon segments were fixed, paraffin-embedded, sectioned, and subjected to either H&E staining for histological scoring.

## H&E staining and histopathological analysis

Mouse colon specimens were fixed with 4% formaldehyde for 12 hours before dehydration with gradient alcohol. The samples were then cleared with xylene, embedded in paraffin, and cut into 5 μm thick sections. The tissue sections were stained with H&E. The H&E stains were scored blinded by a pathologist based on inflammatory cell infiltration, hemorrhage, and epithelium disruption on a scale of 0 to 3 (mild to severe).

## Statistics and reproducibility

Data are presented as mean ± standard deviation (SD) for biochemical experiments and mean ± standard error of the mean (SEM) for pathological experiments. The number of the sample size (n) and statistical hypothesis testing method are described in the legends of the corresponding figures. Statistical analyses of data were performed with GraphPad Prism v9.3 or OriginPro v8.5. Experiments in Fig. 1f, Fig. 4a, b, Fig. 5d, e, and Supplementary Fig. 10a, b have been repeated at least twice with similar results.

## Reporting summary

Further information on research design is available in the Nature Portfolio Reporting Summary linked to this article.

## Data availability

The atomic coordinates for the $TcsH-TMPRSS2^{ECD}$ complex and $TcsH^{2236-2618}-TMPRSS2^{ECD}$ complex have been deposited in the Protein Data Bank (PDB) under the accession code 8JHZ and 8JI0, respectively. The EM maps of the $TcsH-TMPRSS2^{ECD}$ complex and $TcsH^{2236-2618}-TMPRSS2^{ECD}$ complex have been deposited in the Electron Microscopy Data Bank (EMDB) with the accession codes EMD-36301, and EMD-36303, respectively. The local EM map of the C-terminal part of the $TcsH-TMPRSS2^{ECD}$ complex has been deposited in the EMDB with accession code EMD-36302. All other data are available from the corresponding author upon reasonable request. Source data are provided with this paper.

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

## Acknowledgements

We would like to thank Dr. Yigong Shi for the guidance and discussion. We also appreciate the technical support from the Biomedical Research Core Facility, High-Performance Computing Center, and Laboratory Animal Resources Center at Westlake University. This study was partially supported by the National Key R&D Program of China (Grant no. 2023YFC2308403 to L.T.), "Pioneer" and "Leading Goose" R&D Program of Zhejiang (Grant no. 2024SSYS0029), the Zhejiang Provincial Natural Science Foundation of China (Grant no. LR20C010001 to L.T.), and the Westlake Center for Genome Editing (Program no. 21200000A992210 to L.T.). L.T. also acknowledges support from the Westlake Laboratory of Life Sciences and Biomedicine and the Westlake Education Foundation.

## Author contributions

L.T. conceived the project and designed the experiments. R.Z., L.H., and J.Z. performed microbiological, cell biological, and biochemical assays. L.H. performed animal experiments. R.Z., L.H., J.Z., X.Zhang, Y.L., and X.Zhan performed protein expression and purification, cryo-EM structure determination, and structural data analysis. R.Z., X.Zhan, and L.T. wrote the manuscript with input from all co-authors.

## Competing interests

The authors declare no competing interests.
