## [Peer Review File · Nature Communications]

REVIEWER COMMENTS

Reviewer #1 (Remarks to the Author):

This paper describes the molecular recognition of TMPRSS2 and Hemorrhagic toxin (TcsH), a major virulence factor produced by *Paenibacillus sordellii*. Transmembrane Serine Protease 2 (TMPRSS2) was identified as a host receptor of TcsH. The author determines the cryo-EM structure of the TcsH-TMPRSS2 complex and finds that TcsH binds to the serine protease domain (SPD) of TMPRSS2 via the CROP unit-VI.

The complex structure information is essential and very interesting to broad readers of Nature Communications, though some major and minor points should be revised.

Major points

The author finally confirmed that TMPRSS2 activity was hindered by TcsH binding domain.

In contrast, it is interesting whether some inhibitor of TMPRSS2 inhibits the interaction between toxin and receptor (TMPRSS2). This is crucial information. Please add this information.

In introductions (160-62), low pH inside the endosome triggers a conformational change of DRBD. Is there any evidence of making pores in TcsH or related enzymes? Please add information about this. The author stated that the complex structure is the one at neutral pH; thus, what is the conformational change induced by acidic endosome? Please add the final summary figure: the successive figures of the scheme.

Minor points

Please describe the difference between TcsH and TcsL. Please add the sequence alignment of these toxins as a supplementary figure, including TcdA and TcdB.

Fig1a

In legend, SR and LR should be stated. Otherwise, it is no information about blue and red in CROPs. What is the difference between SR and LR? Please add this information.

1f

Pulldown assay: The band of TMPRSS2 seems to be double or triple. On the other hand, TMPRSS2ECD seems stable based on purified enzyme CBB staining (Supp Fig.2).

Fig3

3a and 3b figures should be enlarged from fig2c using the box.

Fig5b

Please make sure the relationship between a and b.

Reviewer #2 (Remarks to the Author):

Hemorrhagic toxin (TcsH) and TcsL are major virulence factors produced by *P. sordellii*. Recently, transmembrane serine protease 2 (TMPRSS2) was identified as a host receptor of TcsH. This manuscript reported two cryoEM structures of the TcdH-TMPRSS2 complexes: a low resolution structure of the full length TcsH in complex with TMPRSS2-ECD, and a 3 angstrom structure of the TcsH (2236-2618)-TMPRSS2-ECD complex. The interactions involved the serine protease domain (SPD) of TMPRSS2 and the CROP unit-VI of TcsH, and the structural findings were validated by site-specific mutagenesis using pull down and a cell-based binding assay. It was also reported that the binding of TcsH inhibited the proteolytic activity of TMPRSS2, while the underlying physiological relevance remains unknown. While an interesting topic, this manuscript seems to be put together in a rush, leaving many important questions remain untouched. At the current form, it's more like a pure structural paper, which needs to be significantly strengthened by relevant functional studies.

Major concerns:

1. What's the cytotoxicity of the TMPRSS2-binding deficient TcsH mutants or the E/K mutants in LR4-6? Will K to E mutations in LR1-3 affect glycan binding, cell binding, and toxicity?
2. Is there any relevant animal disease model to validate the structural findings reported here? This is crucial.
3. Why is it a single band for the "pulldown" samples in Fig. 4b, but two bands in sFig. 9? Why was the critical F2521I mutant not included/tested in Fig. 4b and 4c? Importantly, why is this single F to I mutation so powerful considering that the interface is in fact not small? How about a reversed I to F mutant on TcdA?
4. Line 196, this cell-based binding is more than just mediated by fucosylated glycans. I don't think the conclusion in this section is solid based on this single assay. Just an example, what will happen if the K in LR1-3 is changed to E? This is an interesting topic, worthy of spending more time here.
5. What did they learn from the in silico docking? Does it explain why some LRs bind glycans, but others don't? Does it support the E/K hypothesis?
6. Following the same line, their in silico docking results appeared to contradict their earlier finding. They need to seriously consider this instead of conveniently blaming on a yet unknown receptor (p12).
7. Line 206-219: this assay should be repeated using some of the TMPRSS2-binding deficient TcsH mutants.
8. Related to the other manuscript on TcsL submitted by the same group, will InsP6 regulate the structure and function of TcsH, and why?

Minor comments:

1. Is there a particular reason to describe the TcsH-TMPRSS2 interface as “complicated” (p7-8)? How big/small is the buried interface?
2. How was the hydrophobicity calculated in Fig. 3?
3. Around lines 170-175, the figure numbers are all wrong. There is no Fig. 4d or 4e.
4. Line 175, “fucosylated glycans” came from nowhere. Some background info is absolutely needed here.
5. Fig. 4a, what was the prey/bait? Why are there so many bands? Fig. 4b-c, there must be some typos as the residue numbers don't match in these two panels.
6. Line 239: do all TcdB2 variants bind TFPI? Another TFPI paper from the Dong lab should be cited too.
7. Why did the authors use fragment 2236-2618 sometimes, but 2210-2618 some other times?

Response to Reviewers (NCOMMS-23-26719-T)

We very much thank the reviewers for their helpful suggestions and comments. Following their suggestions, we have revised our manuscript and made the point-to-point response as below.

Reviewer #1

The complex structure information is essential and very interesting to broad readers of Nature Communications, though some major and minor points should be revised.

Response: We very much appreciate the reviewer's kind support as well as helpful suggestions and comments.

Major points

1) The author finally confirmed that TMPRSS2 activity was hindered by TcsH binding domain. In contrast, it is interesting whether some inhibitor of TMPRSS2 inhibits the interaction between toxin and receptor (TMPRSS2). This is crucial information. Please add this information.

Response: We thank the reviewer for the nice suggestion. Following the suggestion, we have further tested some commercially available serine protease inhibitors of TMPRSS2, including Bromhexine, Nafamostat, and Avoralstat. Nafamostat modifies Ser441 on TMPRSS2 (Bryan et al. 2022), Bromhexine and Avoralstat may also target the catalytic triad of TMPRSS2 (Kailas et al. 2021). None of them protected cells from TcsH. These chemicals may be too small to interfere with this toxin-receptor interaction. (Line 239-244, Figure 6f)

2) In introductions (160-62), low pH inside the endosome triggers a conformational change of DRBD. Is there any evidence of making pores in TcsH or related enzymes? Please add information about this. The author stated that the complex structure is the one at neutral pH; thus, what is the conformational change induced by acidic endosome? Please add the final summary figure: the successive figures of the scheme.

Response: Thank you for the comments. To our knowledge, there's no structural evidence showing that TcsH or other LCTs make pores in the endosome. In fact, people still don't know what the exact conformational change is for LCTs inside the acidic endosome, which is a long-lasting question in the field. We have added the information to the text. We have also added a final summary figure of the scheme as suggested. (Line 264, Supplementary Fig. 11)

Minor points

1) Please describe the difference between TcsH and TcsL. Please add the sequence alignment of these toxins as a supplementary figure, including TcdA and TcdB.

Response: TcsH and TcsL are toxins that belong to the same family with a sequence identity of ~43.4%. These are very large proteins (2364 residues for TcsL and 2618 residues for TcsH) and thus showing the whole sequence alignment for these toxins is not common and would take several pages. Below is a chat showing the sequence identity among reference sequences of TcsH, TcsL, TcdA, TcdB, and Tcn α .

	Sequence identity* (%)				
	TcsH	TcsL	TcdA	TcdB	Tcn α
TcsH	100				
TcsL	43.4	100			
TcdA	77.2	41.3	100		
TcdB	42.9	76.2	41.2	100	
Tcn α	25.0	26.7	24.8	26.6	100

*Reference sequences are TcsH and TcsL from *P. sordellii* VPI9048, TcdA and TcdB from *C. difficile* 630, and Tcn α from *C. novyi* GD211209.

2) *Fig1a. In legend, SR and LR should be stated. Otherwise, it is no information about blue and red in CROPs. What is the difference between SR and LR? Please add this information*

Response: As suggested, we have stated the SR and LR in the Figure legend part. The difference between SR and LR has been described in the Introduction section. (Line 66-68, 775-777)

3) *1f. Pulldown assay: The band of TMPRSS2 seems to be double or triple. On the other hand, TMPRSS2ECD seems stable based on purified enzyme CBB staining (Supp Fig.2).*

Response: The bands shown in Fig. 1f are TcsH fragments but not TMPRSS2^{ECD}. The lower bands are degraded TcsH fragments. To avoid the confusion, we have replaced the blot. We are sorry for the misleading and have modified the figure legend for clarity. (Figure 1f, Line 784-785)

4) *Fig3. 3a and 3b figures should be enlarged from fig2c using the box.*

Response: As suggested by the reviewer, we have modified these panels for clarity. (Figure 3)

Fig5b. Please make sure the relationship between a and b.

Response: Thank you for the suggestion. We have modified the Fig. 5b to better explain the relationship between 5a and 5b. (Figure 5b)

Reviewer #2

While an interesting topic, this manuscript seems to be put together in a rush, leaving many important questions remain untouched. At the current form, it's more like a pure structural paper, which needs to be significantly strengthened by relevant functional studies.

Response: We thank the reviewer for the comments and suggestions. We have performed more relevant functional studies to strengthen our manuscript.

Major concerns:

1) *What's the cytotoxicity of the TMPRSS2-binding deficient TcsH mutants or the E/K mutants in LR4-6? Will K to E mutations in LR1-3 affect glycan binding, cell binding, and toxicity?*

Response: As suggested by the reviewer, we have built the TMPRSS2-binding deficient TcsH mutants, including TcsH^{Q2520A/F2521A}, TcsH^{E2544R}, and TcsH^{F2557N/L2558N}. These toxin mutants showed reduced cytotoxicity to the MCF-7 *GMDS*^{-/-} cells (~20-fold) but not *TMPRSS2*^{-/-} cells. Since E2544R is an effective mutation to disrupt the TMPRSS2-binding based on the cytopathic assay, TcsH^{E2544R} was later chosen for the *in vivo* experiments. (Line 178-183, Figures 4d and 4e)

We have also mutated the K in LR1-3 (TcsH¹⁸³²⁻²²⁰⁹) to E. However, all mutants containing multiple K to E mutations with a K1903E site could hardly be expressed. TcsH^{1832-2209/K1903E} and TcsH^{1832-2209/K2037E/K2171E} can be normally expressed but failed to bind the MCF7 *TMPRSS2*^{-/-} cells. These results suggest that the E residue in LR is critical but not adequate for the recognition of fucosylated glycans, other residues in the sequence are also important. We have added these data to our manuscript and changed the writings in the text. (Line 208-213, Figures 5b and 5d)

2) *Is there any relevant animal disease model to validate the structural findings reported here? This is crucial.*

Response: We thank the reviewer for the important suggestion. As *TMPRSS2* is an intestinal epithelial receptor for TcsH, we have adopted the mouse colon-loop ligation experiment to validate the *TMPRSS2*-TcsH structure *in vivo*. We showed that TcsH^{E2544R} caused less damage to the colonic epithelium compared to TcsH, suggesting that the key mutation disrupting the *TMPRSS2*-TcsH structure affects the pathological effect caused by TcsH *in vivo*. (Line 245-254, Figures 6g and 6h)

3) *Why is it a single band for the "pulldown" samples in Fig. 4b, but two bands in sFig. 9? Why was the critical F2521I mutant not included/tested in Fig. 4b and 4c? Importantly, why is this single F to I mutation so powerful considering that the interface is in fact not small? How about a reversed I to F mutant on TcdA?*

Response: The smaller bands in sFig.9 are degraded fragments of TcsH²²¹⁰⁻²⁶¹⁸, possibly caused by freeze-thawed. Using freshly purified proteins can reduce the degradation. We have changed the blots to avoid the confusion. The F2521I mutation is designed based on the sequence alignment between TcsH and TcdA. We did test this important position in Fig. 4b and 4c (Q2520A/F2521A and Q2520F/F2521R). It seems that F2521 is an important residue contributing to the interaction. As suggested, we have made a reversed I to F mutation (the according position in TcdA is 2613) on TcdA. This reversely mutated residue in TcdA did not render the TcdA CROPs overt binding to *TMPRSS2*. (Line 190-192, Supplementary Fig. 9)

4) *Line 196, this cell-based binding is more than just mediated by fucosylated glycans. I don't think the conclusion in this section is solid based on this single assay. Just an example, what will happen if the K in LR1-3 is changed to E? This is an interesting topic, worthy of spending more time here.*

Response: We thank the reviewer for the insightful comment. As suggested, we have mutated the K in LR1-3 (TcsH¹⁸³²⁻²²⁰⁹) to E. Notably, all mutants containing multiple K to E mutations with a K1903E site could hardly be expressed. TcsH^{1832-2209/K1903E} and TcsH^{1832-2209/K2037E/K2171E} can be normally expressed but failed to bind the MCF7 *TMPRSS2*^{-/-} cells. These results suggest that the E residue in LR is critical but not adequate to recognize fucosylated glycans, other residues in the sequence are also important. We have included the data in the manuscript and changed the writings in the text. (Line 208-213, Figures 5b and 5d)

5) *What did they learn from the in silico docking? Does it explain why some LRs bind glycans, but others don't? Does it support the E/K hypothesis?*

Response: The *in silico* docking provides us with possible positions where the glycans bind, which supports our E/K hypothesis. (Line 219-221)

6) *Following the same line, their in silico docking results appeared to contradict their earlier finding. They need to seriously consider this instead of conveniently blaming on a yet unknown receptor (p12).*

Response: We agree with the reviewer's comment. Here, it is somehow arbitrary to blame on a yet unknown receptor without additional evidence. Therefore, we have modified the sentence in the paragraph. (Line 286-288)

Line 206-219: this assay should be repeated using some of the TMPRSS2-binding deficient TcsH mutants.

Response: Following the reviewer's suggestion, we have performed the experiment with some *TMPRSS2*-binding deficient mutants included as comparisons. (Line 238-243, Figure 6e)

7) *Related to the other manuscript on TcsL submitted by the same group, will InsP6 regulate the structure and function of TcsH, and why?*

Response: The CROPs domain of TcsH also partly regulated the InsP6-induced autocleavage in the extracellular environment. We have related data and discussion in the other manuscript.

Minor comments:

1) *Is there a particular reason to describe the TcsH-TMPRSS2 interface as "complicated" (p7-8)? How big/small is the buried interface?*

Response: Following the reviewer's comment, we have calculated the area of the buried interface using the *dr_sasa* software (Ribeiro, J., 2019) and rephrased the description. The buried interface between the TcsH and *TMPRSS2* is about 1130 Å². (Line 158-160)

2) *How was the hydrophobicity calculated in Fig. 3?*

Response: The hydrophobicity surface calculated in Fig. 3 was performed using the ChimeraX. The coloring is from dark cyan for most hydrophilic through white to dark goldenrod for most hydrophobic.

3) *Around lines 170-175, the figure numbers are all wrong. There is no Fig. 4d or 4e.*

Response: We are sorry for the mislabeling. Fig. 4d should be Fig. 4a, Fig.4a should be Fig.4b, and Fig. 4e should be Fig. 4c. We have corrected these mistakes. (Line 173-178)

4) *Line 175, “fucosylated glycans” came from nowhere. Some background info is absolutely needed here.*

Response: As suggested by the reviewer, we have added some background information here. (Line 88-90, Ref. #37)

5) *Fig. 4a, what was the prey/bait? Why are there so many bands? Fig. 4b-c, there must be some typos as the residue numbers don't match in these two panels.*

Response: In Fig.4a, Fc-tagged Tmprss2^{ECD} and its variants serve as the bait and His-tagged TcsH²²¹⁰⁻²⁶¹⁸ is the prey. The smaller bands are degraded fragments of TcsH²²¹⁰⁻²⁶¹⁸, possibly caused by freeze-thawed. Using freshly purified proteins can reduce the degradation. We have replaced the blots to avoid the confusion. We have also corrected the typos in Fig. 4c. (Figures 4a and 4c)

6) *Line 239: do all TcdB2 variants bind TFPI? Another TFPI paper from the Dong lab should be cited too.*

Response: Based on the structure, TcdB2 sequences have the potential to bind TFPI. However, we did not experimentally test every sequence from various strains. As suggested, we have cited the paper from the Dong lab. (Line 274, Ref. #49)

7) *Why did the authors use fragment 2236-2618 sometimes, but 2210-2618 some other times?*

Response: Sorry for the inconvenience. This is due to an early miscommunication between two collaborating groups. One group made mutations on 2236-2618 while another worked on 2210-2618. We have experimentally confirmed that both fragments bind to Tmprss2 with similar affinity and the data has been included in the supplementary materials. (Supplementary Fig. 4)

REVIEWER COMMENTS

Reviewer #1 (Remarks to the Author):

All of my concerns were addressed properly.

Reviewer #2 (Remarks to the Author):

Major concerns:

1. For all the TcsH mutations discussed in this manuscript, there is zero characterization of proper protein folding. As they pointed out, some K to E mutants could not be expressed properly. For example, did E2544R or other double/triple E/K mutants fold correctly?

BTW, the results of E2544R in Fig. 6g looks not much better than the WT. As the authors suggested that both TMPRSS2 and fucosylated glycans serve as the receptors, I suggest they combine E2544R with their E to K mutant to disrupt both the protein and glycan receptors simultaneously. Will that bring the toxicity closer to the control group?

2. The section of "Recognition of fucosylated glycans by TcsH" is very weak, probably should be deleted if no further supports. This was largely based on E to K mutant in LR4-6 and cell-based binding assay, which was not directly related to fucosylation. Such mutants could simply change the overall charge distribution of this protein, which may not be relevant to glycan binding at all. It would be more informative if they compare these E/K mutants on TMPRSS2^{-/-} and GMDS^{-/-} cells.

3. The in silico docking in this manuscript is more like a wishful thinking without any evaluation of the quality of docking, nor any experimental validation. Furthermore, these results appeared to contradict their earlier finding showing that the ectodomain of TMPRSS2 protected the TMPRSS2^{-/-} cells from TcsH (ref. 35). The authors suggested that there are at least three glycan binding sites. While one is relatively close to (but not clash to) TMPRSS2, the other two are quite far away from it and ready for glycan binding. Therefore, it's unlikely that "the competition is due to steric hindrance between TMPRSS2 and the protein part of fucosylated glycoproteins".

Reviewer #3 (Remarks to the Author):

Peer Review: Molecular basis of TMPRSS2 recognition by Paenibacillus sordellii hemorrhagic toxin

Ruoyu Zhou, Liuqing He, Jiahao Zhang, Xiaofeng Zhang, Yanyan Li, Xiechao Zhan and Liang Tao (2023).

Manuscript No: NCOMMS-23-26719A

General Overview and Comments:

Recent findings have identified Transmembrane Serine Protease 2 (TMPRSS2) as a receptor for the Large Clostridial Toxin (LCT), TcsH. However, the basis for this interaction remained unknown. The work presented here confirms the structure of TcsH in complex with TMPRSS2. Despite the similarity to TcdA from *C. difficile*, this binding to TMPRSS2 is not conserved.

In this revised manuscript,

Key New Results

- Mutation of TcsH at residues expected to facilitate TMPRSS2 binding resulted in a 20 reduction in toxicity via in vitro analysis using MCF7 -GMDS-/- cells, which express TMPRSS2, but fail to express another known TcsH receptor; fucosylated glycans. This toxicity was restored when using cells that lack the expression of TMPRSS2; highlighting the role this protease has in binding TcsH.
- Differences in TcdA and TcsH were examined and investigated for differences that may be involved in carbohydrate binding of these toxins. Mutations to these residues resulted in decreased binding to fucosylated cell surface glycans.
- To assess in TcsH binding to TMPRSS2 resulted in reduced proteolytic activity from this receptor, fluorescent reporter assays were performed. Full length TcsH did repress proteolytic activity from TMPRSS2, but TcsH mutants that lacked the binding region specific for TMPRSS2 failed to block proteolysis. Despite this, use of known TMPRSS2 inhibitors did not sterically hinder TcsH binding and activity, leading to intoxication.
- A comparison of TcsH and a truncated TcsH incapable of TMPRSS2 binding was tested in a colonic loop model for toxicity. In its WT form, TcsH was able to induce colonic damage, including a significant erosion of the colonic epithelium, some oedema and inflammation. The authors suggest there is a significant reduction in the potency of truncated TcsH, however, the truncated TcsH is still able to induce significant epithelial damage, oedema and inflammation.

Conclusions

- TcsH binds to TMPRSS2 via the end of the CROPs domain
- Cryo-EM structures of this interaction will be useful for understanding how LCTs bind to their cognate receptors, and may shed light on novel therapeutic targets

Concerns/Originality

- I question how different the level of colonic damage induced by WT TcsH and TcsHE2544R is (based on the representative image). Based on the images provided, the scores stipulated in Fig 6H (for only 3 test conditions) do not appear to be as significantly different as suggested.
- The mucosal oedema, inflammatory cell infiltration, epithelial disruption, haemorrhage, and overall pathologies scores for the TcsHE2544R toxin appear to be underrepresented based on the image provided.

- The lack of intoxication inhibition by known Tmprss2 inhibitors is also of concern, as it brings into question the validity of the binding specificity, and strengthens the suggestion that TcsHE2544R toxin is still able to induce significant damage (likely through fucosylated glycans).

Suggestions

- Repeat experiments of the colonic loops should be considered as the data provided does not convincingly show a significant reduction in pathology. It is unclear how or who performed the scoring. In any case this should be completely blinded, and in this instance, I would suggest sent to an independent pathologist to assess the extent of the damage.
- As it stands, I do not believe there has been a significant change in the outcome of the work presented here to that presented to the previous reviewers.

Response to Reviewers (NCOMMS-23-26719-B)

We appreciate the reviewers and editors for their comments and suggestions, which help to further improve our manuscript. Following their suggestions, we have revised the manuscript and made the point-to-point response as below.

Reviewer #2

1) *For all the TcsH mutations discussed in this manuscript, there is zero characterization of proper protein folding. As they pointed out, some K to E mutants could not be expressed properly. For example, did E2544R or other double/triple E/K mutants fold correctly?*

Response: We thank the reviewer for the comment. We have performed the circular dichroism (CD) spectra analysis and validated that all full-length TcsH mutants used in our study are properly folded. Moreover, these TcsH mutants have similar cytotoxicity on the *TMPRSS2*^{-/-} cells compared to the WT TcsH, indicating they are functionally active. (Line 182-185, Supplementary Fig. 8)

BTW, the results of E2544R in Fig. 6g looks not much better than the WT. As the authors suggested that both TMPRSS2 and fucosylated glycans serve as the receptors, I suggest they combine E2544R with their E to K mutant to disrupt both the protein and glycan receptors simultaneously. Will that bring the toxicity closer to the control group?

Response: We also noticed that TcsH^{E2544R} caused noticeable epithelial damage in the colon-loop assay, possibly due to the residual recognition of TcsH^{E2544R} to TMPRSS2. We further combined E2544R with Q2520A/F2521A and made a triple-point mutated TcsH (TcsH^{Q2520A/F2521A/E2544R}, or TcsH^{AAR}). By measuring the cytotoxicity on the MCF-7 WT and *TMPRSS2*^{-/-} cells, we demonstrated that this triple-point mutation can better abolish the TMPRSS2-mediated entry compared to E2544R or Q2520A/F2521A. We thus used TcsH^{AAR} to perform the colon-loop assay and showed that this mutant causes obviously reduced damage to the intestinal epithelium compared to the WT toxin. (Line 179-188, 222-230, Fig. 4d, e, Fig. 6)

2) *The section of “Recognition of fucosylated glycans by TcsH” is very weak, probably should be deleted if no further supports. This was largely based on E to K mutant in LR4-6 and cell-based binding assay, which was not directly related to fucosylation. Such mutants could simply change the overall charge distribution of this protein, which may not be relevant to glycan binding at all. It would be more informative if they compare these E/K mutants on TMPRSS2^{-/-} and GMDS^{-/-} cells.*

Response: We thank the reviewer for the comment. We agree that this part is somewhat weak as we did not obtain the co-structure of TcsH-fucosylated glycans. Following the reviewer's and editor's suggestion, we have removed this section from our manuscript.

3) *The in silico docking in this manuscript is more like a wishful thinking without any evaluation of the quality of docking, nor any experimental validation. Furthermore, these results appeared to contradict their earlier finding showing that the ectodomain of TMPRSS2 protected the TMPRSS2^{-/-} cells from TcsH (ref. 35). The authors suggested that there are at least three glycan binding sites. While one is relatively close to (but not clash to) TMRPSS2, the other two are quite far away from it and ready for glycan binding. Therefore, it's unlikely that “the competition is due to steric hindrance between TMPRSS2 and the protein part of fucosylated glycoproteins”.*

Response: We appreciate the reviewer for the comment. We agree that this part could be controversial without further experimental validation. As suggested by the reviewer and editor, we have removed the section “Recognition of fucosylated glycans by TcsH” from the manuscript.

Reviewer #3

1) *I question how different the level of colonic damage induced by WT TcsH and TcsHE2544R is (based on the representative image). Based on the images provided, the scores stipulated in Fig 6H (for only 3 test conditions) do not appear to be as significantly different as suggested.*

The mucosal oedema, inflammatory cell infiltration, epithelial disruption, haemorrhage, and overall pathologies scores for the TcsHE2544R toxin appear to be underrepresented based on the image provided.

Response: We thank the reviewer for pointing out this. It is possibly due to the residual recognition of TcsH^{E2544R} to TMPRSS2 based on the cell intoxication experiment. To address this, we further combined E2544R with Q2520A/F2521A and made a triple-point mutated TcsH (TcsH^{Q2520A/F2521A/E2544R}, or TcsH^{AAR}). By measuring the cytotoxicity on the MCF-7 WT and *TMPRSS2*^{-/-} cells, we demonstrated that this triple-point mutation can better abolish the TMPRSS2-mediated entry compared to E2544R or Q2520A/F2521A. Thus, we used TcsH^{AAR} to replace TcsH^{E2544R} in the colon-loop assay. The result shows that this mutant causes obviously reduced damage to the intestinal epithelium compared to the WT toxin. (Line 179-188, 222-230, Fig. 4d, e, Fig. 6)

2) *The lack of intoxication inhibition by known TMPRSS2 inhibitors is also of concern, as it brings into question the validity of the binding specificity, and strengthens the suggestion that TcsHE2544R toxin is still able to induce significant damage (likely through fucosylated glycans).*

Response: We thank the reviewer for the comment. This assay was previously proposed by another reviewer. Since these inhibitors were initially designed to block the protease activity, it is not surprising that no inhibition for the toxin-binding was detected. While the inhibitors (such as nafamostat) modify the substrate pocket of TMPRSS2 via phenylguanidino acylation (Fraser et al. 2022), TMPRSS2 interacts with TcsH through the surface loops. Below is the overlapped structure showing the phenylguanidino acylation superimposed on the TMPRSS2-TcsH complex. No direct competition between TcsH and phenylguanidino acylation was observed. To reduce the potential confusion, we have moved this figure to the supplementary materials. (Line 217-220, Supplementary Fig. 11b, c)

3) *Repeat experiments of the colonic loops should be considered as the data provided does not convincingly show a significant reduction in pathology. It is unclear how or who performed the scoring. In any case this should be completely blinded, and in this instance, I would suggest sent to an independent pathologist to assess the extent of the damage.*

Response: As suggested by the reviewer, we have increased the number of samples, and the tissue damage was assessed by an independent pathologist blindly. (Line 532-534, Fig. 6b)

REVIEWERS' COMMENTS

Reviewer #2 (Remarks to the Author):

All of my concerns have been properly addressed.

Reviewer #3 (Remarks to the Author):

Based on my previous comments there were concerns regarding the toxicity of TcsH-E2544R, which did not appear to be that different from WT TcsH.

To address these concerns the authors have included a new mutant of TcsH, TcsHQ2520A/F2521A/E2544R, or TcsHAAR, which, when used in colonic loop assays now more closely represents untreated colonic tissue following intoxication.

This, combined with the inclusion of additional repeats of the assay and blind analysis of the data by a pathologist have addressed my concerns, and I am satisfied with the outcome of these experiments.

Concerns regarding the lack of inhibition of intoxication by TMPRSS2 inhibitors has also been sufficiently addressed.

I have no further concerns that require addressing and do not require further revisions prior to publication